# Ensemble Model Based on Hybrid Deep Learning for Intrusion Detection in Smart Grid Networks

**DOI:** 10.3390/s23177464

**Published:** 2023-08-28

**Authors:** Ulaa AlHaddad, Abdullah Basuhail, Maher Khemakhem, Fathy Elbouraey Eassa, Kamal Jambi

**Affiliations:** Department of Computer Science, Faculty of Computing and Information Technology, King Abdulaziz University (KAU), Jeddah 21589, Saudi Arabia; makhemakhem@kau.edu.sa (M.K.); feassa@kau.edu.sa (F.E.E.); kjambi@kau.edu.sa (K.J.)

**Keywords:** Smart Grid, deep learning, intrusion detection, distributed denial of service attacks, communication infrastructure, real-time monitoring

## Abstract

The Smart Grid aims to enhance the electric grid’s reliability, safety, and efficiency by utilizing digital information and control technologies. Real-time analysis and state estimation methods are crucial for ensuring proper control implementation. However, the reliance of Smart Grid systems on communication networks makes them vulnerable to cyberattacks, posing a significant risk to grid reliability. To mitigate such threats, efficient intrusion detection and prevention systems are essential. This paper proposes a hybrid deep-learning approach to detect distributed denial-of-service attacks on the Smart Grid’s communication infrastructure. Our method combines the convolutional neural network and recurrent gated unit algorithms. Two datasets were employed: The Intrusion Detection System dataset from the Canadian Institute for Cybersecurity and a custom dataset generated using the Omnet++ simulator. We also developed a real-time monitoring Kafka-based dashboard to facilitate attack surveillance and resilience. Experimental and simulation results demonstrate that our proposed approach achieves a high accuracy rate of 99.86%.

## 1. Introduction

The Smart Grid, powered by digital information and control technologies, offers immense potential to transform the traditional electric grid into a more reliable, secure, and efficient system. The Smart Grid enables real-time analysis and precise control by integrating advanced communication networks and state estimation techniques, leading to optimized energy distribution and improved grid resilience. However, the increasing dependence on interconnected communication networks also exposes the Smart Grid to cyber threats, jeopardizing its reliability and functionality [1,2,3,4,5]. Electric utilities all over the world use SCADA (supervisory control and data acquisition) protocols. Those protocols are often used in Smart Grid operations to measure parameters, monitor processes, and control operations with measurement and control systems [3]. The electric network’s SCADA system is essential [6]. It comprises computer systems that talk to each other and share important information across networks. The widespread adoption of IT has made these systems susceptible to hacking attempts [5]. Therefore, the development of effective intrusion detection and prevention systems has become paramount to safeguarding the networks against such attacks [7,8,9].

Incorporating intrusion detection enables the detection of potential threats both before and after they infiltrate a system. The most effective method for integrating the gateway with an IEC 61850-based network is to implement it internally within the gateway [10]. IEC 61850 does not mandate any particular method for detecting attacks or repairing damage if it occurs; nevertheless, an intrusion detection system (IDS) might be used inside the grid to bolster IEC 61850’s security [11]. The prevalence of possible threats in the electric infrastructure grows with the rise of machine-to-machine (M2M) and human−machine interface (HMI) communication [12,13]. IDS is crucial for the safety of the Smart Grid. The radio channel used for data transmission in the Smart Grid network is also susceptible to cyberattacks, as it is the foundation of the entire network. Hence, intrusion detection in a Smart Grid’s SCADA network is a hot topic in cyber security research [8,14].

On the other hand, distributed generation (DG) has been the key to transitioning to renewable energy sources (RES). When DG is introduced at different nodes in an existing network, it changes the overall shape of the power grid. Changes in voltage and current at individual nodes result in more points of entry into the power grid [15]. The scope of an electrical network directly impacts the complexity of the communication technologies and supporting infrastructure that makes up the Smart Grid as a whole. Some researchers have been motivated to examine the cyber dangers to Smart Grids after considering the limitations of existing intelligent grid communication systems. In Figure 1 we present Smart Grid arch with current attacks and future attacks in distributed systems.

This paper proposes a novel hybrid deep-learning method for detecting DDoS attacks on the Smart Grid’s communication infrastructure. Our approach combines the power of convolutional neural networks (CNNs) and recurrent gated units (RGUs) to analyze network traffic patterns and identify anomalous behaviors indicative of DDoS attacks. By leveraging the strengths of these two techniques, we aim to enhance the accuracy and efficiency of intrusion detection in the Smart Grid context. We employ two distinct datasets to evaluate the performance of our proposed method. The first dataset is obtained from the Intrusion Detection System dataset provided by the Canadian Institute for Cybersecurity, which comprises a wide range of network attack scenarios. Additionally, we generate a custom dataset using the Omnet++ simulator, enabling us to simulate realistic Smart Grid network environments and incorporate specific attack scenarios. By utilizing these diverse datasets, we aim to evaluate the effectiveness and robustness of our approach under various attack scenarios.

Furthermore, recognizing the importance of real-time monitoring and situational awareness in the Smart Grid, we developed a comprehensive dashboard that provides visualization and monitoring capabilities. This dashboard enables operators to monitor the network’s health, detect ongoing attacks, and facilitate timely response and mitigation. The results of our experiments demonstrate the superior performance of the proposed hybrid deep-learning method by achieving an accuracy rate of 99.86% in detecting DDoS attacks. This result outperforms existing intrusion detection systems and highlights the potential of deep-learning techniques in fortifying the Smart Grid’s cybersecurity defenses.

Our contributions are as follows:Presenting a method that combines a hybrid model with gated recurrent unit (GRU) and convolutional neural network (CNN) to prevent DDoS attacks in the smart-energy-grid distributed system.We utilized DDoS Evaluation Dataset (CIC-DDoS2019) datasets for in-depth research evaluation, demonstrating that the proposed approach outperformed existing IDS algorithms with a 99.86% accuracy and nearly 100% detection rate.Developing a Kafka-based dashboard system that enhances resilience to attacks and provides real-time monitoring capabilities for network administrators.Generating a custom dataset through simulations that incorporate various Smart Grid devices, including both attack traffic and normal traffic.

The remainder of the paper is structured as follows:

Section 2: Literature Review—This section provides an overview of the literature and research on DDoS attacks in the Smart-Grid domain. It examines the strengths and limitations of previous approaches and sets the foundation for the proposed hybrid algorithm.

Section 3: Proposed Hybrid Algorithm—In this section, the suggested hybrid algorithm, which combines a convolutional neural network (CNN) and a recurrent gated unit (GRU), is explained in detail. The architecture, training methodology, and critical components of the algorithm are discussed, emphasizing how it addresses the challenges of DDoS attacks in the Smart Grid.

Section 4: Performance Evaluation and Simulation—Simulations are conducted to evaluate the performance of the proposed algorithm. A comprehensive comparison is made with existing methods, including accuracy, detection, and false-positive rates. The results are analyzed and discussed, highlighting the superiority of the proposed algorithm.

Section 5: Conclusion and Future Work—This section provides concluding remarks on the findings and contributions of the research. It summarizes the key insights from the study and discusses potential avenues for future research and improvements in addressing DDoS attacks in the Smart Grid domain.

## 2. Related Work

A security plan must be implemented to safeguard the intelligent grid [16]. The researchers in [17] proposed several monitoring procedures to keep an eye out for suspicious branch flow variations and anomalous load deviations to identify fake data injection (FDI) attacks. In this work [18], the authors proposed detecting FDI attacks using deep learning. The proposed method for finding FDI is based on a warning system made using custom metrics.

### 2.1. Firewall and Intrusion and Protection

An individualized firewall designed by [19] was created based on the idea of the SCADA Wall, which was designed to protect SCADA networks powered by Comprehensive Packet Inspection (CPI) technology. It addresses the limitations of traditional SCADA firewalls by offering a deeper payload inspection, enhanced protection for proprietary industrial protocols through the Proprietary Industrial Protocols Extension Algorithm (PIPEA), and abnormality detection within industrial operations using the Out-of-Sequence Detection Algorithm (OSDA). A comparative analysis with two commercial SCADA firewalls demonstrates the effectiveness of SCADA Wall in mitigating the drawbacks without compromising on the low latency requirement of SCADA systems.

The authors of [20] presented DIDEROT (Dnp3 Intrusion Detection prevention system), an Intrusion Detection and Prevention System (IDPS) specifically designed for DNP3 SCADA systems. It combines supervised machine learning and unsupervised/outlier machine learning models to identify DNP3 cyberattacks and anomalies. The system first utilizes a supervised ML model to detect specific cyberattacks, and if the network flow is deemed normal, an unsupervised/outlier ML model is activated to detect possible anomalies. The DIDEROT performance is demonstrated using real data from a substation environment, and it leverages Software Defined Networking (SDN) technology for timely mitigation of detected cyberattacks and anomalies.

The research paper [21] proposes a parallel structure using Recurrent Neural Network (RNN) classifier models, specifically Long Short-Term Memory (LSTM) and Gated Recurrent Units (GRU), for improved detection accuracy. The model is trained and tested using a dataset created from an experimentally generated SDN-based SCADA topology. Transfer learning is employed to enhance the model’s performance further. An additional 5% improvement was achieved through transfer learning. The findings indicate that the proposed RNN deep-learning classifier model effectively detects DDoS attacks in SDN-based SCADA systems. The researchers [22] created a cyber-physical monitoring system to find infiltration and DoS in the smart meter. The idea only works when online events and real-world facts are combined in a way that makes sense. The test demonstrates that the model can effectively identify threats by connecting cyber and physical signals.

It has been suggested in [23] that cyber-attack breaches in SCADA systems could be tracked by looking for patterns in time. In addition, the method was developed to track any odd shifts in the functioning of the linked system. The artificial neural network (ANN) method and a hidden Markov model were used to build the model in order to achieve this goal. Five feature extraction methods were used to test how well the suggested model worked in simulations and real-world situations. Satisfactory results were found from the strategy executed using the time-feature extraction model termed MAGPIE [24], which is a novel smart home intrusion detection system that autonomously adjusts its anomaly classification models based on changing conditions within the smart home environment. MAGPIE utilizes a probabilistic cluster-based reward mechanism and non-stationary multi-armed bandit reinforcement learning to adapt its decision function. The system incorporates both cyber and physical data sources and detects human presence to optimize accuracy. The experimental evaluation conducted in a real household demonstrates the high accuracy of MAGPIE. The open-source availability of MAGPIE and its evaluation datasets allows for future advancements and the integration of additional data sources as smart home environments and attacks evolve.

In order to provide a security model for the SCADA system that can be used in gas and oil facilities, it has been suggested that a C4.5 decision tree algorithm should be used [25]. However, the openness of the SCADA network also poses security risks, including cyber-attacks and information security breaches.

The authors of [26] presented a SCADA system testbed for cybersecurity research, focusing on a water storage tank’s control system. The study included conducting sophisticated cyber-attacks and training machine learning algorithms to detect these attacks using captured network traffic data. The trained models are then deployed in the network for real-time attack detection. The results highlight the effectiveness of the machine learning models in detecting attacks in SCADA environments. Overall, this research provided valuable insights into SCADA system cybersecurity and offered practical implications for enhancing security measures. It revealed that 99.8% of the F1 test successfully identified the attacks.

### 2.2. Performance Enhancement and Evaluation

The authors of [27] focused on enhancing the SCADA system’s resilience against DDoS attacks using three machine learning algorithms: J48, Naive Bayes, and Random Forest. The algorithms were trained and evaluated using the KDDCup’99 dataset, and preprocessing techniques were applied. The results reveal that the Random Forest classifier achieved the highest accuracy rate of 99.99%, while the Naive Bayes classifier performed slightly lower at 97.74%. This research contributed valuable insights into improving the effectiveness of machine learning algorithms for detecting attack patterns in SCADA systems, providing a foundation for enhancing the security of critical infrastructures. The researchers in [28] focused on securing Smart Grid networks through intrusion detection systems (IDS). Compared with traditional machine learning techniques, the study investigated the performance of ensemble learning techniques, specifically bagging-based, boosting-based, and stacking-based. The evaluation is based on critical metrics such as detection rate, false alarm rate, miss detection rate, and accuracy using the CICDDoS 2019 benchmark dataset. The results demonstrate that the stacking-based ensemble learning techniques outperformed other algorithms across all of the evaluation metrics. This research contributes valuable insights into improving the effectiveness of IDS for securing Smart Grid networks.

### 2.3. Data Confidentiality Mechanisms and Defenses

Software Defined Networking (SDN) [29] provided a way to find and stop DDoS attacks. This technique was based on discrete wavelet transforms and auto-encoder neural networks. Wavelet transforms were used to pull out statistical features, which were then used by an auto-encoder neural network to identify DDoS attack samples. The authors in [29] proposed [30] a novel feature selection−whale optimization algorithm−deep neural network (FS-WOA–DNN) method that uses a novel approach combining feature selection using whale optimization and deep neural network techniques to counter distributed denial-of-service attacks. In order to enhance the security of the proposed methodology, the researchers homomorphically encrypt standard data before uploading it to the cloud, ensuring complete confidentiality. The experiment’s results demonstrated an accuracy of 95.35 % when identifying DDoS attacks.

In this paper [31], the authors proposed a novel multi-scale residual classifier (MSRC) method for detecting network traffic anomalies. The approach involves dividing the traffic into subsequences with different observation scales, utilizing wavelet transform to extract time−frequency information, employing a stacked automatic encoder (SAE) to learn data distribution, calculating reconstruction error vectors, and leveraging the multipath residual group to capture feature information across scales. The experimental results demonstrate that the proposed method outperforms traditional approaches for detecting abnormal network traffic. The findings highlight the significance of incorporating extensive observation and multiple transformation scales to uncover diverse information within network traffic. Overall, this research contributes to advancing network anomaly detection by considering multi-scale characteristics and achieving improved detection performance.

### 2.4. Machine Learning and Deep-Learning Techniques

In response to these concerns, the authors of [32] propose a token authentication service module as a defense mechanism against distributed denial-of-service (DDoS) attacks. The effectiveness of the proposed security defense architecture is demonstrated through a simulated experiment conducted in an energy management system. The experimental results validate the capability of the proposed architecture to enhance security and its compatibility with real-world field systems.

The authors of [33] proposed a solution utilizing sFlow and adaptive polling for sampling, along with integrating Snort Intrusion Detection System (IDS) and a deep-learning model based on Stacked Autoencoders (SAE). Leveraging the flexibility of Software Defined Networking (SDN), the approach allows for network programming without relying on third-party hardware or software. The proposed system demonstrates higher detection accuracy through performance metrics and evaluation, achieving a True Positive rate of 95% with a False Positive rate below 4% in the sFlow implementation compared with adaptive polling. This research enhances DDoS attack detection in IoT networks, leveraging the benefits of SDN and deep-learning techniques.

The authors of [34] evaluated the performance of two open-source intrusion detection systems (IDSs), Snort and Suricata, for accurately detecting malicious traffic on computer networks. The study further explored a hybrid version of SVM and Fuzzy logic, which yielded improved detection accuracy. Nevertheless, the best results were achieved by employing an optimized SVM with the firefly algorithm, achieving a false-positive rate (FPR) of 8.6% and a false-negative rate (FNR) of 2.2%. This outcome indicates a significant performance improvement. The novelty of this work lies in its comparison of the two IDSs at a high network speed of 10 Gbps and the application of hybrid and optimized machine learning algorithms to enhance Snort’s functionality.

When sorting and finding things, the model was more accurate when it used a back propagation neural network. However, the backpropagation neural network algorithm proposed by [35], whose primary job was to find threats to the resources, had a low rate of finding attacks. The algorithm’s main task was to identify resource threats. This paper [36] addressed the issue of network intrusions threatening the data security of the train Ethernet Consist Network (ECN), which is responsible for transmitting critical train control instructions. To counter these threats, the authors proposed a novel ensemble intrusion detection method tailored to defend against various network attacks, including IP Scan, Port Scan, Denial of Service (DoS), and Man in the Middle (MITM). The method achieved a high detection performance, with an accuracy rate of 0.975.

The study by [37] focused on enhancing the performance of the k-Nearest Neighbor (kNN) algorithm for classifying botnet attacks in the IoT environment. The proposed method achieved the highest accuracy and fastest execution time among the evaluated techniques. This paper [38] highlighted applying deep-learning (DL) and machine learning (ML) techniques, specifically a hybrid framework called HCRNNIDS, for predicting and classifying malicious cyberattacks in networks. HCRNNIDS combines the strengths of convolutional neural networks (CNN) and recurrent neural networks (RNN) to capture both local and temporal features, resulting in an improved performance and prediction accuracy. With a high detection rate accuracy of up to 97.75%, HCRNNIDS offers a promising solution for effectively identifying and mitigating network threats.

The authors of [39] introduced the SCDNN model, which combines spectral clustering (SC) and deep neural networks (DNN) for intrusion detection in complex network datasets. The results highlight the superiority of the SCDNN model in terms of accuracy and robustness, highlighting its potential for detecting and mitigating network intrusions more effectively than conventional methods.

This paper [40] highlighted the critical role of Intrusion Detection Systems (IDSs) in securing networks and computer systems by utilizing artificial intelligence (AI) techniques, specifically deep-learning algorithms such as Convolutional Neural Networks (CNNs). Overall, this survey paper provides valuable insights into the application of CNNs in IDSs, addressing the current state of research and identifying areas for future exploration and improvement.

## 3. Background

### 3.1. Deep Learning

Deep learning is a rapidly evolving field within artificial intelligence (AI) that has gained significant attention in recent years. It uses neural networks with multiple layers to automatically learn and extract complex patterns and representations from raw data. One of the distinguishing features of deep learning is its ability to bypass the need for manual feature selection [41]. Traditionally, in machine learning, experts manually engineer relevant features to train models. However, deep-learning models can automatically learn and extract meaningful features directly from raw input data, eliminating the laborious and time-consuming feature engineering process. It is a field that encompasses various definitions put forth by researchers. Despite variations, common keywords and concepts emerge, including “complex architectural data model”, “unsupervised machine learning”, “learning multiple layers”, and “nonlinear data transformations”. Deep-learning models typically consist of multiple layers, allowing for hierarchical learning of representations. Each layer in the network performs specific computations, progressively building up a hierarchy of features. This hierarchical architecture enables the network to capture low-level and high-level abstractions, thus capturing intricate patterns and structures within the data.

Several deep-learning architectures have been developed to tackle different data types and tasks. Examples include Deep Belief Networks (DBNs), which are generative models capable of unsupervised learning; Recurrent Neural Networks (RNNs), which excel in processing sequential data; and Convolutional Neural Networks (CNNs), which have proven to be highly effective in tasks involving structured grid-like data such as images and audio. The versatility and power of deep learning have sparked tremendous interest and research efforts. Researchers continue to explore and develop innovative deep-learning techniques, addressing challenges such as model interpretability, training on limited labeled data, and improving the efficiency of deep-learning algorithms.

### 3.2. Convolutional Neural Network (CNN) for Intrusion Detection

Convolutional neural networks (CNNs) have emerged as a powerful tool in the field of intrusion detection, particularly in the context of securing Smart Grids. The increasing complexity and interconnectedness of Smart Grid systems have made them vulnerable to various cyber threats, including malicious attacks. Robust intrusion detection systems (IDSs) are essential to detect and mitigate these threats.

CNNs offer a promising approach for intrusion detection in Smart Grids because they can automatically learn and extract relevant features from raw input data. In the context of Smart Grids, these input data may include network traffic data, sensor readings, and communication patterns. By leveraging the inherent hierarchical architecture of CNNs, these models can capture local and global patterns in the data, enabling accurate and efficient detection of intrusions. Applying CNNs in Smart Grid intrusion detection involves designing appropriate network architectures to process Smart Grid systems’ complex and dynamic data effectively. Convolutional layers in the CNN are responsible for learning and detecting local features within the input data, such as abnormal patterns in network traffic or unusual behaviors in sensor readings. Pooling layers downsample the feature maps, allowing the model to focus on the most relevant and informative features. Finally, fully connected layers connect these learned features to the output layer, where the intrusion detection decision is made.

The use of CNNs in Smart Grid intrusion detection systems offers several advantages:They can manage high-dimensional and heterogeneous data, making them well-suited for the diverse data sources found in Smart Grids.CNNs can adapt to changes in the data patterns over time, allowing them to detect evolving intrusion techniques effectively.By automatically learning features from the data, CNN-based IDSs can reduce the dependence on manual feature engineering, which can be time consuming and error prone.

Developing and optimizing CNN architectures for Smart Grid intrusion detection is an active area of research. Researchers are exploring different network configurations, hyperparameter settings, and training strategies to improve the detection accuracy and efficiency of CNN-based IDSs in Smart Grid environments. Additionally, efforts are being made to integrate CNNs with other machine learning techniques, such as recurrent neural networks (RNNs), to capture temporal dependencies in the data and enhance intrusion detection capabilities. CNNs offer a promising approach for intrusion detection in Smart Grids, leveraging their ability to automatically learn and extract relevant features from complex and heterogeneous data. By applying CNNs in the context of Smart Grid intrusion detection, researchers aim to enhance the security and resilience of Smart Grid systems against various cyber threats. The ongoing advancements in CNN architectures and techniques hold great potential for improving intrusion detection’s accuracy, efficiency, and effectiveness in the evolving landscape of Smart Grids.

### 3.3. Intrusion Detection Using Gated Recurrent Unit (GRU)

Intrusion detection is critical to securing computer networks and systems against malicious activities. Traditional approaches to intrusion detection often rely on rule-based or statistical methods. However, with network traffic’s increasing complexity and dynamic nature, more advanced techniques are required to detect and mitigate intrusions effectively. One such technique is GRU.

GRU is a type of recurrent neural network (RNN) that has gained significant attention in intrusion detection due to its ability to model and capture sequential dependencies in data. Unlike traditional RNNs, GRU introduces gating mechanisms that enable it to update and forget information selectively over time. This capability allows GRU to effectively manage long-term dependencies in the data, which is crucial for detecting complex intrusion patterns. In intrusion detection, GRU models are trained on network traffic data, system logs, or other relevant data sources to learn the typical patterns and behaviors of the network. These models can then detect anomalies or deviations from the learned normal behavior, which may indicate a potential intrusion or malicious activity.

The architecture of a GRU-based intrusion detection system typically consists of multiple layers of GRU units. Each unit processes sequential data at a specific time step and passes the hidden state to the next time step. The secret state retains essential information from previous time steps, allowing the model to capture long-term dependencies. The output of the GRU units can be fed into a classification layer, which makes the final intrusion detection decision based on the learned representations.

GRU-based intrusion detection systems offer several advantages:They can effectively model and capture temporal dependencies in network traffic data, allowing for more accurate intrusion detection than traditional methods.GRU models can adapt to changing network behaviors and detect emerging intrusion patterns, making them suitable for dynamic and evolving network environments.GRU’s gating mechanisms enable them to efficiently process long data sequences, making them well-suited for real-time intrusion detection.

However, the performance of GRU-based intrusion detection systems heavily depends on various factors, such as the quality and representativeness of the training data, hyperparameter tuning, and model optimization. Researchers are actively investigating techniques to enhance the performance and robustness of GRU models for intrusion detection. This includes exploring different network architectures, incorporating attention mechanisms, and further integrating other deep-learning techniques to improve intrusion detection′s accuracy and efficiency. GRU has emerged as a powerful tool for intrusion detection, particularly in scenarios involving sequential data, such as network traffic. Its ability to capture long-term dependencies and adapt to changing network behaviors makes it a valuable asset in detecting complex intrusion patterns. Ongoing research and advancements in GRU-based intrusion detection aim to enhance the effectiveness and reliability of detecting intrusions in dynamic and evolving network environments.

### 3.4. Artificial Neural Networks Structure

An artificial neural network is an initiative-taking system comprising many closely connected, nonlinear processing units, parallel, or “devices” that are very good at computing. Instead, it may be seen as a collection of flexible mathematical structures that can identify complex nonlinear connections between the datasets that are being input and those that are being produced [42]. A typical neural network is made up of many elementary processing units that are coupled and known as neurons. Each neuron produces a series of activations that have real-world values. Sensors in the environment turn on input neurons layer, convolutional layer, pooling layer and GRU layer neurons connected with output layer (see Figure 2).

Mathematical Representation

As inputs to the proposed model, the first layer of GRU is responsible for processing the inputs and generating the outputs. The outputs from the first layer are transferred into the system for the second layer. Similarly, the outputs of the second layer serve as inputs for the third layer. Using an activation function allows for the generation of the GRU model’s final outputs. We use two activation functions that are most often utilized, namely the sigmoid and tanh, which are presented as examples in the work [43,44,45].

Next, we show the mathematical formulation of our proposed method.

Input vector:

Let $X=\left[X_{1},\;X_{2},\;\ldots ,\;X_{N}\right]T$ denote the input vector, where each $X_{i}$ represents a specific feature or measurement in the Smart Grid network.

Gated Recurrent Unit (GRU) for feature extraction:

The GRU layers are employed to extract relevant features from the input vector. The GRU gates and activation function are defined as follows:(1)$$Z_{t}=\sigma \left(W_{z\left[X_{t},\;H_{\left\{t-1\right\}}\right]}+B_{z}\right)$$
(2)$$R_{t}=\sigma \left(W_{r\left[X_{t},\;H_{\left\{t-1\right\}}\right]}+B_{r}\right)\;$$
(3)$$H_{t}^{\prime }=\tanh\left(W_{h\left[X_{t},\;R_{t}*H_{\left\{t-1\right\}}\right]}+B_{h}\right)\;$$
(4)$$H_{t}=\left(1-Z_{t}\right)*H_{\left\{t-1\right\}}+Z_{t}*H_{t}^{\prime }$$

Here, $Z_{t}\;$ represents the update gate, $R_{t}$ is the reset gate, $H_{t}^{\prime }$ is the candidate activation, $H_{t}$ denotes the output of the GRU unit at time t, and $X_{t}$ corresponds to the input at time t. $W_{z}$, $W_{r}$, $W_{h}$, $B_{z}$, $B_{r}$, and $B_{h}$ are the weight matrices and bias vectors of GRU. The matrix transpose operation is denoted by the superscript “T”.

Convolutional Layers:

The extracted features from the GRU layers are passed through a series of convolutional layers for further processing. Let $H_{1}$, $H_{2}$, $H_{3}$, and $H_{4}$ denote the hidden feature maps obtained after each convolutional layer, respectively:(5)$$H_{1}=ConvBlock_{1\left(H_{t}\right)}\;$$
(6)$$H_{2}=ConvBlock_{2\left(H_{1}\right)}\;$$
(7)$$H_{3}=ConvBlock_{3\left(H_{2}\right)}\;$$
(8)$$H_{4}=ConvBlock_{4\left(H_{3}\right)}\;$$

$ConvBlock_{i}$ functions represent the operations performed within each convolutional layer.

Pooling and Flattening:

The feature maps obtained from the convolutional layers undergo pooling and flattening. The average pooling operation is applied to summarize the features, and the flattened output is denoted as F:(9)$$P=AvgPooling\left(H_{4}\right)\;$$
(10)$$F=Flatten\left(P\right)\;$$

Fusion and Classification:

The features from the GRU layers ($H_{t}$) and the flattened features (F) are concatenated to capture the combined information:(11)$$C=Concatenate\left(H_{t},\;F\right)\;$$

Average pooling:(12)$$\Psi^{n}=P_{\left(avg\right)\left(\Psi^{n-1}\right)}$$
where:

$\Psi^{n}$ represents the pooled feature map at level n.

$P_{\left(avg\right)}$ is the average pooling function.

Flattening layer:(13)$$L=flatten\left(h^{4}\right)\;$$
where:

L denotes the flattened representation.

$h^{4}$ is the output of the fourth convolutional layer.

Concatenation of GRU and CNN outputs:(14)$$c{\;}^{t}=concate\left(K,\;L\right)$$

Softmax function for classification:(15)y^z_i=e^z_i / ∑n=1^c_t e^z_n
where:

$\hat{y}\left(z\right)\_i$ represents the *i*-th element of the predicted output vector using the Softmax function.

z_i is the *i*-th element of the input vector.

c_t represents the total number of classes.

Cross-entropy loss function:(16)E_pl=−1/b ∑_i=1^n y_i log_2y^_i 
where:

$E\_p\left(l\right)\;$ denotes the cross-entropy loss.

b represents the batch size.

y_i is the true label of the *i-*th sample.

$\hat{y}\_i$ is the predicted label of the *i-*th sample using the Softmax function.

The concatenated features (C) are then passed through fully connected layers and an appropriate activation function for classification purposes.

Output and Loss Function:

The final output of the model, denoted as y_pred, represents the predicted class probabilities. The choice of activation function and loss function depends on the specific classification task and requirements of the Smart Grid intrusion detection system.

## 4. Proposed Hybrid Model

The proposed hybrid deep-learning method for the intrusion detection system that can be seen in Figure 3 includes both CNN and GRU models. This option was chosen because it is widely believed that CNN is superior to other methods for accurately capturing position-invariant properties. The GRU module keeps track of long-term dependencies and uses memory cells to obtain valuable information from the collected data. The reset gate is used to delete or remove useless data. Several factors affected the GRU model choice. In order to further deepen the network, the algorithmic architecture was outfitted with three GRU blocks and four CNN blocks. The main goal of the convolution layer is to perform its namesake function so that a feature map can be made from the input data by extracting the features. The convolutional kernel in a convolutional network multiplies the input data. A nonlinear function then sets off the network. This action is performed inside the convolutional network to capture feature mapping. The convolution kernel randomly determines the weights and biases [46]. After the completion of each CNN layer comes the addition of a normalization layer and a max pooling layer. A “pooling operation” determines the most significant or typical value for all of the characteristics in a particular area’s nearby vicinity.

The GRUs’ output and the CNNs’ output are merged in the concatenation layer, which also receives the output of the CNN layers after it has been flattened. After the concatenation layer, two layers that were already linked are joined together. In order to avoid overfitting, a dropout layer is added after the last layer that is ultimately linked. The SoftMax layer, connected to the classification layer, maps the output to a probability distribution. This enables the classification layer to make accurate predictions about the kinds of labels. We evaluated this algorithm by applying it to the NSL-KDD99 and local generated dataset and compared the results by using hybrid model of CNN and the GRU deep-learning model. The hybrid model outperformed in terms of accuracy, precision, detection rate, false-positive rate (FPR), and F1 score. However, the researchers in [47] argued that the NSL-KDD99 dataset is no longer valid. They mentioned that since the network traffic in that dataset was generated in 1998, it may not fully represent current network topologies and attack dynamics. Therefore, we intend to utilize the CIC-DDOS2019 cyber security dataset with the algorithm. This dataset contains a wide range of contemporary attack scenarios, enabling the simulation of real-world conditions.

### 4.1. Details of Dataset

The CIC-DDoS2019 dataset [34] comprises 50,063,112 records, with 50,006,249 rows corresponding to DDoS attacks and 56,863 rows representing benign traffic. Each row in the dataset contains 86 features. Table 1 provides an overview of the attack types in the CIC-DDoS2019 dataset and a brief description of each attack.

The training dataset includes 12 different DDoS attacks, namely, Domain Name System (DNS), Network Time Protocol (NTP), Lightweight Directory Access Protocol (LDAP), Network Basic Input Output System (NetBIOS), Microsoft SQL Server (MSSQL), Simple Network Management Protocol (SNMP), User Datagram Protocol (UDP), Simple Service Discovery Protocol (SSDP), UDP-Lag, SYN, WebDDoS, and TFTP. On the other hand, the test dataset contains seven attacks, including MSSQL, NetBIOS, PortScan, LDAP, UDP, UDP-Lag, and SYN, observed during the testing day. Each attack type represents a specific DDoS attack methodology. For instance, the NTP-based attack utilizes the reflection technique to flood a target with increased UDP traffic by exploiting Network Time Protocol servers. Similarly, the other attack types leverage various network protocols or vulnerabilities to launch DDoS attacks.

Overall, the CIC-DDoS2019 dataset provides a comprehensive collection of records capturing DDoS attacks and benign traffic, enabling the evaluation of intrusion detection models for Smart Grid networks. Creating specialized datasets facilitates the analysis of machine learning and deep-learning approaches in binary and multi-class classification scenarios, allowing for a thorough assessment of their efficiency in tackling various attack types.

As illustrated before, Table 1 presents the characteristics of the dataset used. The initial stages in the preparation stage include replacing not a number (NaN), cleaning of data, and endless fields with the mean value of the column. This is completed before moving on to the next stage. The characteristics are then transformed into numerical features and included in the dataset with any other numerical features that may already be present. In addition, the labels in the dataset go through a process that turns them into numbers. This makes the label “harmless” equal to 0 and “DDoS” equal to 1. In order to reduce the number of distinct feature variations, the dataset was normalized and mapped out similarly. [0, 1] denotes the range of the uniform mapping interval. Figure 4 shows that the correlation matrix was not used in this research because the dataset had no extraneous qualities or features related to each other. Because of this, each of the accessible characteristics affected how the model made decisions.

The outcomes of normalization on the dataset are shown in the list in Table 1. Due to the normalizing process, all character attributes have been changed into the numbers that correspond with them. The dataset was then divided into a training set and a testing set, with a ratio of 66:33 between the two sets. The remaining 33 % of the data were used for validation and testing once the first training was completed, utilizing 66 % of the data.

The four essential features that come together to form the confusion matrix are the ones that are used to set the measurement parameters for the classifier. 

The following are some of them: The term “true positive”, abbreviated as “TP”, refers to the correct forecast made by an algorithm. In addition, a “true negative”, shortened to “TN”, is an accurate but pessimistic prediction made by the algorithm. “FP” stands for “false positive”, which is when an algorithm predicts a positive class even though the actual class is negative. A label is said to be falsely negative (sometimes abbreviated as FN) if an algorithm predicts that it would be negative, but it turns out to be positive. The performance metrics that judge how well an algorithm works are its Accuracy, Precision, Recall, and F1score.
(17)$$Accuracy=\frac{TP+TN}{TP+TN+FP+FN}\;$$
(18)$$Precision=\frac{TP+TN}{TP+FP}\;$$
(19)$$Recall=\frac{TP+TN}{TP+FN}\;$$
(20)$$F1score=\frac{2\left(precision\times recall\right)}{precision\times recall}\;$$

### 4.2. Determining Convolutional Layers and GRU Units

The architecture of a hybrid deep-learning model significantly influences its performance and generalization capabilities. We employed a systematic and iterative approach to determine the optimal number of convolutional layers and GRU Units for our hybrid model.

Convolutional Layers: A model′s number of convolutional layers impacts its ability to extract spatial features from the input data. We initiated our experimentation with a baseline architecture and gradually increased the number of convolutional layers while keeping other hyperparameters constant. Through this process, we monitored the model′s performance on the validation set, focusing on accuracy, precision, recall, and F1score metrics. We aimed to strike a balance between model complexity and performance improvement. When we observed diminishing returns or signs of overfitting, we selected the configuration that demonstrated the most promising trade-off.GRU Units: Similarly, the number of GRU units in our model′s GRU layers influenced its capacity to capture temporal patterns within sequential data. We began with a baseline configuration and iteratively adjusted the number of GRU units. Our objective was to identify the point at which increasing the number of units led to diminishing returns in performance. By evaluating the model′s performance on the training and validation sets, we ensured that the chosen configuration effectively captured temporal dependencies without overfitting.

Our methodology for determining the number of convolutional layers and GRU units aimed to optimize the hybrid model′s architecture for intrusion detection within Smart Grid networks. It considered the complexities of the data and the potential trade-offs between model complexity and performance gains.

## 5. Experiment Details

In this section, we present comprehensive implementation details of our research to ensure the reproducibility and transparency of our experiments. Our primary objective was to develop a robust and efficient intrusion detection system (IDS) tailored specifically for Smart Grid networks. The primary objective of our research was to develop a robust and efficient intrusion detection system (IDS) explicitly tailored for Smart Grid networks. To achieve this, we proposed a novel ensemble learning approach that combines the power of deep-learning techniques, such as CNN and GRU networks. The hybrid deep-learning model was trained and evaluated using a comprehensive dataset comprising real-world Smart Grid network traffic and a simulated intrusion scenarios custom dataset. We conducted extensive experiments to assess the model’s accuracy, precision, recall, and F1-score performance. Additionally, various parameters, including learning rates, batch sizes, and network architectures, were fine tuned to optimize the model’s efficiency. The results of our experiments demonstrated the superiority of our proposed approach over traditional methods, showcasing its potential to enhance the security and resilience of smart energy grids against malicious cyber-attacks.

### 5.1. Custom-Based Dataset Generation

To create a realistic and diverse dataset for resilient, intelligent grid attack detection, we employed a multi-step methodology leveraging OMNeT++ and various other tools and technologies. This process encompassed the following key steps:Design the Smart Grid Topology: We defined the network topology, representing the various components of the Smart Grid, including power plants, substations, smart meters, and communication infrastructure.Model Normal Traffic: We simulated normal traffic patterns within the Smart Grid network, incorporating power consumption, data exchange, and communication protocols. These simulations accounted for variations in traffic volume, timing, and network conditions.Identify Attack Scenarios: We identified a set of attack scenarios based on established Smart Grid attack vectors, such as Denial of Service (DoS), false data injection, tampering with meter readings, and compromising communication channels.Define Attack Parameters: The characteristics of each attack scenario were specified, encompassing attack type, duration, intensity, and targeted components. These parameters were diversified to ensure the dataset’s richness.Select Deep-Learning Algorithms: Suitable deep-learning algorithms, including CNN and GRU, were chosen based on their proven performance in intrusion detection tasks.Generate Deep-Learning Predictions: The chosen deep-learning algorithms were applied to the simulation data, generating ensemble predictions for normal and attack traffic instances. This step contributed to capturing diverse attack patterns for enhanced detection accuracy.Collect Simulation Data: Network traffic, communication logs, sensor readings, and system events were collected during simulations for normal and attack scenarios.Feature Extraction: Relevant features were extracted from the collected data, including packet attributes, communication patterns, energy consumption, and network performance metrics.Labeling and Augmentation: Instances were labelled appropriately based on simulation conditions (normal or attack). Data augmentation techniques were employed to introduce feature variations, enhancing dataset diversity.Split the Dataset: The generated dataset was divided into training, validation, and testing sets to assess the performance of the ensemble-learning-based intrusion detection model.Train Ensemble Learning Models: The training set was used to train ensemble learning models by optimizing multiple base learners’ parameters to achieve high detection accuracy.Performance Evaluation: The performance of ensemble-learning models was evaluated using validation and testing sets, with metrics such as accuracy, precision, recall, and F1-score.Fine Tuning and Optimization: Models were fine-tuned based on evaluation outcomes to enhance their robustness, sensitivity, and specificity when detecting Smart Grid attacks.

The above approach facilitated the creation of a comprehensive and realistic custom dataset for resilient Smart Grid attack detection, enhancing the authenticity and effectiveness of the training and evaluation process.

The proposed method enabled the creation of a comprehensive and realistic custom dataset for resilient Smart Grid attack detection. Combining OMNeT++ simulation, other tools and technologies, attack-scenario design, and ensemble-learning techniques enhanced the dataset’s variability, accuracy, and generalization capabilities. By incorporating diverse attack patterns and utilizing ensemble predictions, the generated dataset facilitated the development and evaluation intrusion detection systems that can effectively mitigate Smart Grid attacks.

#### 5.1.1. Omnet++ Simulation for Dataset Generation

To evaluate the performance of the intrusion detection system (IDS) in the context of a Smart Grid, a custom dataset was generated using the OMNeT++ simulation framework. Using a custom dataset allowed us to emulate realistic scenarios and capture various types of network traffic and attacks specific to Smart Grids. OMNeT++ provides a powerful platform for simulating network behavior and interactions. It allowed us to model the communication infrastructure of a Smart Grid, including the different components such as smart meters, data aggregators, control centers, and communication channels. We could generate various data packets, network flows, and system events by simulating the network environment. In the process of generating the custom dataset, we considered various factors that influence the behavior of a Smart Grid network. These factors included the number of devices, communication protocols, data transfer rates, network topologies, and attacks. By manipulating these parameters, we created realistic network traffic scenarios encompassing normal system operations and different attack scenarios. The dataset generation process involved the creation of different attack scenarios, such as Denial of Service (DoS) attacks, Distributed Denial of Service (DDoS) attacks, replay attacks, injection attacks, and data manipulation attacks. Each attack scenario was carefully designed to reflect the techniques and strategies commonly employed by attackers targeting Smart Grid networks. Furthermore, the dataset included a variety of network traffic patterns, such as periodic data exchanges, event-driven communication, and command−response interactions. This ensured that IDS could effectively capture and analyze different traffic patterns and identify deviations from normal behavior.

To enhance the diversity and realism of the dataset, we incorporated variations in the network conditions and system parameters. This included simulating fluctuations in network bandwidth, latency, packet loss rates, and changes in the Smart Grid’s operational state and load conditions. Considering these factors, we aimed to create a dataset representative of real-world Smart Grid environments. The generated custom dataset serves as a valuable resource for training and evaluating the performance of the IDS for detecting and mitigating intrusions within Smart Grids. It provides a comprehensive set of network traffic samples and attack scenarios that can be used to validate the effectiveness and robustness of the intrusion detection algorithms and techniques employed in the system. In summary, the custom dataset generated using the OMNeT++ simulation framework allows for evaluating and validating the Smart Grid intrusion detection system. It enables the testing of the system’s capability to detect and respond to various types of attacks and network anomalies, thereby enhancing the security and resilience of Smart Grid infrastructures.

#### 5.1.2. Attacking Tools-Based Dataset

To assess the effectiveness of the intrusion detection system (IDS) in detecting and mitigating Distributed Denial of Service (DDoS) attacks in a real-world scenario, a custom dataset was generated using DDoS attack tools. The utilization of such tools allowed for the replication of DDoS attack patterns and for the creation of realistic attack scenarios for evaluation purposes. DDoS attacks pose a significant threat to the availability and reliability of network services, making them a critical concern for intrusion detection systems. We generated a custom dataset focusing on DDoS attacks to enhance the IDS’s capability to accurately identify and respond to these types of attacks in our research context. We selected widely used DDoS attack tools from the cybersecurity community to create this custom dataset. These tools facilitated the simulation of real-world attacker behavior and enabled the initiation and control of DDoS attacks. The attack tools were chosen based on their proven effectiveness, versatility, and compatibility with our target environment.

The custom dataset encompassed various DDoS attacks, including, but not limited to, ICMP Flood, UDP Flood, SYN Flood, HTTP Flood, and DNS Amplification attacks. Each attack type was carefully configured with appropriate parameters to emulate different attack intensities, traffic patterns, and vectors. This allowed for a comprehensive evaluation of IDS’s ability to detect and mitigate diverse DDoS attack scenarios. During the dataset generation process, considerations were given to factors such as attack duration, attack volume, attack sources, and attack variations. By manipulating these factors, we created a dataset that closely resembled real-world DDoS attacks, capturing the complexity and diversity of attack patterns commonly observed in network environments. We also included normal network traffic alongside the DDoS attacks to ensure the dataset’s realism. This combination of normal and attack traffic provided a representative environment for evaluating IDS’s ability to distinguish between legitimate traffic and malicious activities during an attack. It enabled the system to learn and adapt to the unique signatures and characteristics associated with DDoS attacks, improving its accuracy and reducing false positives. The generated custom dataset using DDoS attack tools serves as a valuable resource for training, testing, and evaluating the performance of IDS in detecting and mitigating DDoS attacks.

### 5.2. Suricata with Kafka and Garafana Dashboard

In order to enhance the resilience of smart energy grids and improve real-time monitoring capabilities for intrusion detection, we integrated Suricata with Kafka and utilized the Grafana Dashboard. This combination of tools and technologies provided a comprehensive solution for effective intrusion detection in Smart Grid networks. Suricata is an open-source intrusion detection and prevention system that offers robust network traffic analysis and threat detection capabilities. By deploying Suricata within the Smart Grid environment, we could continuously monitor the network traffic for potential intrusions and attacks. Suricata employs a variety of detection mechanisms, including signature-based detection, protocol analysis, and anomaly detection, to identify malicious activities within the network. To facilitate real-time monitoring and analysis of the network traffic data generated by Suricata, we leveraged Kafka as a distributed streaming platform. Kafka enables the collection, storage, and processing of high volumes of streaming data in a scalable and fault-tolerant manner. By integrating Suricata with Kafka, we ensured that the network traffic data were efficiently captured and made available for further analysis, as shown in Figure 5. To visualize the collected network traffic data and gain actionable insights, we utilized the Grafana Dashboard. Grafana is a powerful data visualization and analytics platform that offers a wide range of customizable dashboards and visualizations. By leveraging Grafana, we were able to create a user-friendly and intuitive interface to monitor the network traffic in real time. The dashboard provides detailed information about network activities, identified intrusions, and potential security threats, allowing operators to promptly respond to any anomalies or attacks, as shown in Figure 6.

The integration of Suricata with Kafka and the utilization of the Grafana Dashboard offers several benefits for intrusion detection in Smart Grid networks. Firstly, it enables real-time monitoring of network traffic, allowing for quick detection and response to potential intrusions or attacks. Secondly, the scalability and fault-tolerance provided by Kafka ensure that the network traffic data are reliably collected and available for analysis. Lastly, the Grafana Dashboard offers a visually appealing and easily interpretable interface to monitor the network traffic and identify any security-related events. By employing Suricata with Kafka and utilizing the Grafana Dashboard, our proposed approach enhanced the resilience of smart energy grids by enabling efficient intrusion detection and real-time monitoring. The integration of these tools facilitated the timely identification of potential threats, ensuring the security and reliability of Smart Grid networks.

### 5.3. Integration of SLIPS with CNN and GRU for Real-Time Intrusion Detection

In this section, we delve into the integration of the StratosphereLinuxIPS (SLIPS), a cutting-edge Behavioral Machine Learning-Based Intrusion Prevention System, with Convolutional Neural Networks (CNN) and Gated Recurrent Units (GRU) to achieve real-time and accurate intrusion detection. SLIPS has been designed to target malicious behaviors within network traffic, focusing on identifying targeted attacks and command and control channels. Notably, SLIPS offers a comprehensive visualization tool that aids analysts in comprehending network security status effectively. The system operates in real time, analyzing diverse data inputs, including live network traffic, pcap files, and network flows generated by tools such as Suricata, Zeek/Bro, and Argus. Leveraging its core behavioral analysis framework, SLIPS processes these input streams, identifying and highlighting anomalous behaviors that warrant immediate attention from security analysts. Through its synergy with advanced machine learning techniques such as CNN and GRU, SLIPS enhances its capabilities to swiftly identify and respond to evolving threats, further solidifying its role as a robust intrusion prevention system. This integration exemplifies the continual evolution of intrusion detection mechanisms to address the ever-evolving landscape of network security challenges, which are presented in Figure 6.

### 5.4. Experiment Environment

Our experiments were conducted in a controlled environment to ensure consistency and repeatability.

#### 5.4.1. Hardware

Our experiments were conducted on a Dell PowerEdge R760 server with an Nvidia A100 GPU. The server’s robust computational capabilities, coupled with the cutting-edge performance of the Nvidia A100 GPU, allowed us to process complex deep-learning computations efficiently. The GPU’s parallel processing prowess greatly expedited the training and evaluation of our hybrid model, enabling us to handle the intricate tasks associated with intrusion detection within Smart Grid networks.

#### 5.4.2. Software

We utilized two prominent deep-learning frameworks, TensorFlow and PyTorch, to construct, train, and evaluate our hybrid deep-learning model. These frameworks offered extensive libraries and tools tailored for designing and optimizing neural networks. TensorFlow and PyTorch empowered us to implement intricate architectures and customize model components to suit the specific requirements of our intrusion detection system. In addition to the deep-learning frameworks, we employed OMNeT++ version 5.7 for simulating the Smart Grid network environment during custom dataset generation. OMNeT++ provided a robust simulation framework that allowed us to replicate real-world network behaviors, interactions, and attack scenarios within a controlled environment. This facilitated the creation of a diverse and comprehensive dataset, enhancing the authenticity of our experiments. Additionally, the OMNeT++ simulation framework was employed for custom dataset generation, as previously described in Section 5.1.1.

## 6. Results

In this section, we show a complete list of the simulation results with our proposed method. We describe our suggested algorithm performance and compare it to some of its primary competitors, such as CNN and GRU.

To assess the impact of different hyperparameter configurations on the performance of the CNN−GRU intrusion detection model, we conducted a series of experiments using various values for key hyperparameters, as shown in Table 2. The following hyperparameters were considered: learning rate, number of convolutional layers, number of GRU units, dropout rate, batch size, and number of epochs. We observed that the choice of learning rate significantly impacted the convergence and overall performance of the model. Higher learning rates resulted in faster convergence, but increased the risk of overshooting the optimal solution. A learning rate of 0.001 consistently yielded a good performance across different datasets. The number of convolutional layers played a crucial role in capturing hierarchical features from the input data. Increasing the number of layers beyond two led to diminishing returns in terms of performance. Therefore, we recommend using two to three convolutional layers for optimal results. The number of GRU units determined the complexity of the temporal modeling. We found that a moderate number of units, such as 32, 64, or 128, balanced capturing temporal dependencies and avoided overfitting. Applying dropout regularization helped mitigate overfitting and improved the model generalization ability. Dropout rates between 0.2 and 0.5 yielded favorable results. The choice of batch size impacted the convergence speed and memory consumption. Larger batch sizes accelerated training, but required more memory. We found batch sizes ranging from 16 to 128 suitable for our experiments. Finally, the number of epochs influenced the model’s training duration and convergence stability. Training for 50 to 100 epochs generally yielded a good performance, but exceeding the optimal number of epochs risked overfitting.

By carefully tuning these hyperparameters, we achieved improved performance and generalization capabilities for our CNN−GRU-based intrusion detection model. The optimal hyperparameter configuration provided an accuracy of 99.86, precision of 99.5%, recall of 99.83%, and F1 score of 99.68% on the test dataset. These results demonstrate the importance of selecting appropriate hyperparameters for effective intrusion detection in the Smart Grid environment.

A heat map, as shown in Figure 7, is a powerful tool for analyzing and visualizing complex multivariate datasets. It is a two-dimensional matrix presented as a picture, enabling a comprehensive understanding of the data. Heat maps specifically highlight correlations among numerical variables, allowing for the identification of patterns and deviations from the norm. By assigning color coding to the correlation matrix, a heat map aids in selecting attributes that are most influential in machine learning models. The correlation matrix depicts the relationships between variables on a spectrum ranging from highly positive to strongly negative. This representation is achieved by assigning colors to individual cells, each representing a specific measurement at a particular distance from the starting point. Through its visual representation, a heat map facilitates the interpretation of extensive datasets, employing various colors to convey a wide range of values within the tabular format of the two-dimensional data.

The input features, such as the destination port, forward header length, flow bytes, sub-flow forward packet, minimum packet length, active mean, packet length mean, average packet size, active max, packet length variance, ideal mean, and ideal max are shown in a heat map in Figure 7, which illustrates the correlation matrix between the target variable and the input features.

In order to evaluate the performance of the proposed algorithm, additional simulations were conducted using the hyperparameter values listed in Table 2. Remarkably, the suggested algorithm demonstrated a superior ability to converge towards a solution compared with the other algorithms under comparison. Specifically, the method achieved an outstanding validation performance of 0.01435 at epoch 290, representing the highest attainable validation performance. Among the evaluated algorithms, the GRU algorithm exhibited the best performance, achieving a validation performance of 0.01960 at the 132th epoch. Notably, the CNN method also outperformed the LSTM algorithm, reaching a significant validation performance of 0.026267 at the 157th epoch, whereas the LSTM algorithm attained its best validation performance of 0.02990 at the 42th epoch. It is important to note that both algorithms were evaluated on the same dataset, allowing for a fair and direct comparison of their performance (see Figure 8).

### 6.1. Ablation Study Results and Insights

To comprehensively assess the contributions of individual modules within our proposed CNN−GRU hybrid model, we conducted an ablation study involving three distinct scenarios: the Full CNN−GRU Hybrid Model (Baseline), the CNN-Only Model, and the GRU-Only Model. Each scenario was evaluated on a common dataset, and performance metrics, including accuracy, precision, recall, and F1-score, were utilized to quantify the impact of each module.

a.Full CNN−GRU Hybrid Model (Baseline)

The Full CNN−GRU Hybrid Model, representing our complete proposed architecture, demonstrated a robust performance across all of the evaluation metrics. This scenario served as the benchmark against which the effectiveness of the individual components was measured. The hybrid model exhibited a balanced ability to capture spatial and temporal patterns within the data, yielding a favorable overall detection accuracy. The full CNN and GRU hybrid model results are presented in Figure 9.

b.CNN-Only Model

The CNN-Only Model, where the GRU module was excluded, provided valuable insights into the contribution of the convolutional neural network component. Interestingly, this scenario exhibited a strong performance in terms of precision, highlighting its effectiveness in correctly identifying positive instances, particularly true intrusion cases. However, there was a noticeable reduction in recall and F1-score, indicating a potential limitation in capturing the long-term temporal dependencies characteristic of specific attack patterns.

c.GRU-Only Model

The GRU-Only Model, isolating the gated recurrent unit component, highlighted a different facet of the model’s performance. This scenario excelled in capturing temporal patterns and dependencies within the data, leading to higher recall rates than the CNN-Only Model. However, precision was relatively lower, suggesting a higher likelihood of false positives. This could be attributed to the GRU’s ability to capture nuances in temporal behavior, which may occasionally lead to misclassification.

d.Comparative Analysis

Comparing the scenarios, we observed that the hybrid model’s performance struck a balance between the strengths of the individual modules. While the CNN module emphasized accurately detecting spatial features, the GRU module emphasized temporal aspects, as seen in Figure 10. The ablation study highlighted the constructive collaboration between these components in the hybrid model, resulting in the ability to capture spatial and temporal patterns effectively. This will achieve a higher accuracy and a balanced F1score, indicating a well-rounded intrusion detection capability.

The results of the ablation study offer insights that can guide the design and optimization of future intrusion detection systems. The CNN module excels in capturing immediate spatial signatures of specific attack types, making it suitable for quick response scenarios. On the other hand, the GRU module’s proficiency in recognizing temporal behavior suggests its utility in detecting attacks that manifest over a longer duration. Our ablation study underscores the value of the hybrid approach, highlighting the interplay between spatial and temporal pattern recognition. The balanced performance achieved by the hybrid model underlines its potential to offer enhanced intrusion detection capabilities in complex Smart Grid networks.

Figure 10 demonstrates the confusion matrices that show how well each algorithm performs. The evaluation of the algorithms is conducted with the use of a confusion matrix, with the parameters being things such as accuracy, precision, recall, and the rate of false positives. Figure 11 below represents the evaluation matrices of CNN and GRU. Figure 12 represents normal traffic detection rate and attack detection rate for the CNN and GRU model.

Figure 13 compares the proposed method overall accuracy, precision, recall, and  F1score to those of the other algorithms. The simulation results reveal that the suggested method achieved an accuracy of 99.7%, a precision of 98.1%, a recall of 99.9%, and an F1score of 98.9%, respectively. GRU all attained an accuracy of 98.6%, a precision of 99.5%, a recall of 97.4%, and an F1score of 98.5%. CNN had a recall rate of 97.3%, an accuracy rate of 98.5%, a precision rate of 99.8%, and a score of 98.5% on the F1score scale. The LSTM achieved an accuracy rate of 98.5%, a precision rate of 99.9%, a recall rate of 97%, and an F1score of 98% FPR. In every category except for recall, the suggested model performed much better than the other algorithms. The algorithm’s strong focus on generating false positives (FP) was the reason behind the decrease in recall achieved with the suggested method. The higher number of FP contributed to the denominator in the recall calculation, leading to a decreased recall value.

### 6.2. Comparison with State-of-the-Art Models

The comparison of our proposed model’s performance against existing state-of-the-art techniques, including the CNN and RNN-based ensemble model [36] and the machine-learning-based ensemble model [38], reveals its remarkable capabilities in intrusion detection. Notably, our model outperformed its competitors regarding true-positive rate (TPR), achieving consistently higher scores across all of the evaluated metrics. Specifically, our model achieved an exceptional TPR of 100%. Furthermore, our model demonstrated competitive results in accuracy (ACC) and F1score compared with the leading models. While there is room for improvement in the false-positive rate, our model’s ACC was high enough and highlights its high level of performance as mentioned in Table 3.

In the case of the CIC-DDOS2019 datasets, our model achieved the highest TPR of 99.96% and 100%, respectively, as shown in Table 2. These outcomes underscore our model’s efficacy in detecting Denial of Service (DoS) and Distributed Denial of Service (DDoS) attacks across benchmark and custom datasets. Achieving a high TPR is vital for accurately detecting these attacks, and our model’s consistent performance across multiple datasets underscores its robustness and effectiveness.

## 7. Conclusions

This paper presented a novel approach for intrusion detection in a smart energy grid using a hybrid CNN−GRU-based deep learning algorithm. Integrating Suricata and Kafka facilitated real-time monitoring and flow collection, enabling parallel analysis with the deep-learning model for attack surveillance. Additionally, Grafana was utilized as a visualization tool to provide a comprehensive view of network activity and enhance resilience. The proposed hybrid CNN−GRU model demonstrated an exceptional performance in detecting and classifying intrusions. By leveraging the power of deep learning, the model achieved a high accuracy rate of 99.86%, precision of 99.5%, recall of 99.3%, and F1score of 99.2% on the test dataset. These results signify the effectiveness of the hybrid model in accurately identifying malicious activities and providing reliable security measures for the smart energy grid. Integrating Suricata and Kafka allowed for real-time monitoring and immediate response to potential threats. This combination enhanced the system’s ability to detect and mitigate attacks promptly, thereby ensuring the resilience of the smart energy grid against intrusion attempts. Kafka and Grafana, as flow collectors and visualization tools, enabled comprehensive and intuitive network monitoring. They provided a centralized view of network traffic, facilitating the identification of any anomalies or suspicious patterns. The combination of real-time tracking, deep-learning-based intrusion detection, and visualization through Grafana created a robust and resilient defense mechanism for the smart energy grid. Overall, the hybrid CNN−GRU-based deep-learning algorithm, integrated with Suricata, Kafka, and Grafana, highlighted a remarkable performance in intrusion detection for the smart energy grid. This approach ensured real-time monitoring and attack surveillance and provided the necessary resilience to safeguard critical infrastructure. The findings of this study contribute to the advancement of intrusion detection systems in the context of smart energy grids and pave the way for further research and development in securing the evolving landscape of energy distribution systems.

Our study has introduced a pioneering hybrid CNN−GRU model that highlights encouraging outcomes for detecting DoS/DDoS attacks within Smart Grid systems. However, this is only the starting point, and exciting avenues exist in intrusion detection.

## 8. Future Directions

One promising avenue for future research lies in harnessing the power of ensemble-learning techniques. These techniques involve the collaboration of multiple models, each contributing its unique strengths to create a more robust and accurate intrusion detection system. By amalgamating the capabilities of architectures such as Long Short-Term Memory, Bidirectional Recurrent Neural Networks (BRNN), and Convolutional Neural Networks within an ensemble framework, we envision a heightened understanding of intricate patterns residing in Smart Grid network traffic. Extending our study to encompass Smart Grids—a cornerstone of modern energy distribution systems—is a natural progression. Smart Grids are vulnerable to cyber threats, including DDoS attacks, and adapting our hybrid model to this unique environment is a compelling future step. By embracing ensemble-learning techniques in this context, we aim to bolster the precision and dependability of anomaly detection within critical infrastructure networks.

The essential aspect of our research will be the advancement of explainable AI techniques. As deep-learning models gain traction in critical systems, the need for comprehensible decisions becomes paramount. Striving to unravel the decision-making process and provide meaningful insights into the detection process will elevate the practicality of our proposed system in real-world scenarios. Our study has laid the groundwork for further exploration in the intricate realm of intrusion detection for SCADA and Smart Grid environments. Through the convergence of ensemble learning, expansion to Smart Grids, and heightened interpretability, we endeavor to bolster the cybersecurity of critical infrastructures and combat the ever-evolving landscape of cyber threats. Stay tuned as we embark on this exciting journey of discovery and innovation.

## Figures and Tables

**Figure 1 sensors-23-07464-f001:**
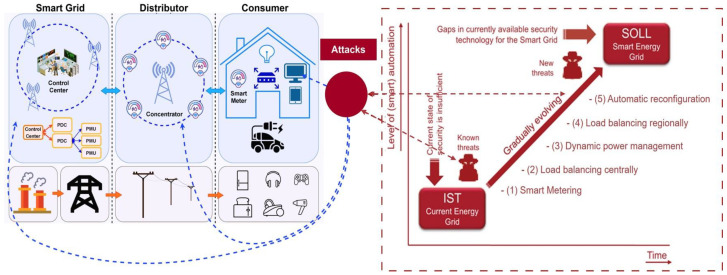
Attacks on Smart-Grid-distributed systems.

**Figure 2 sensors-23-07464-f002:**
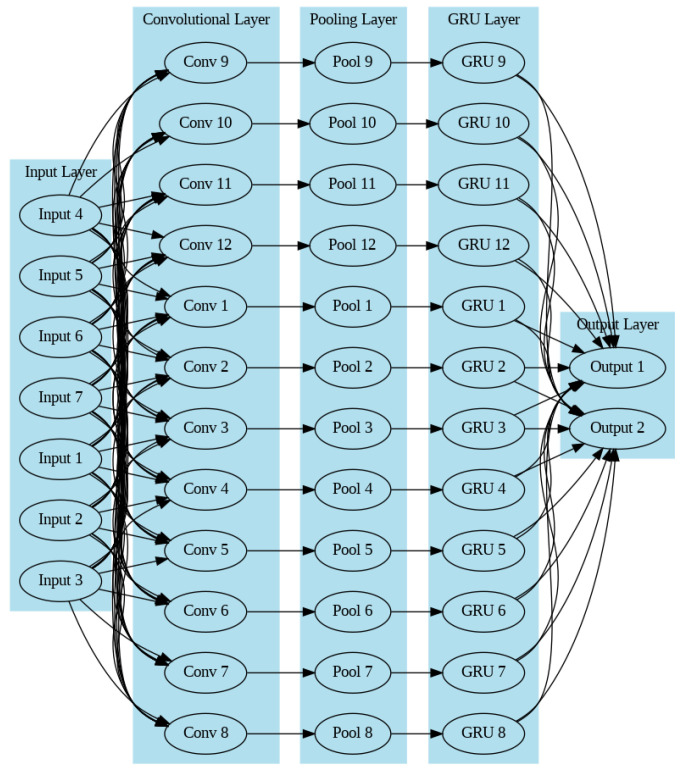
Multilayer perceptron diagram.

**Figure 3 sensors-23-07464-f003:**
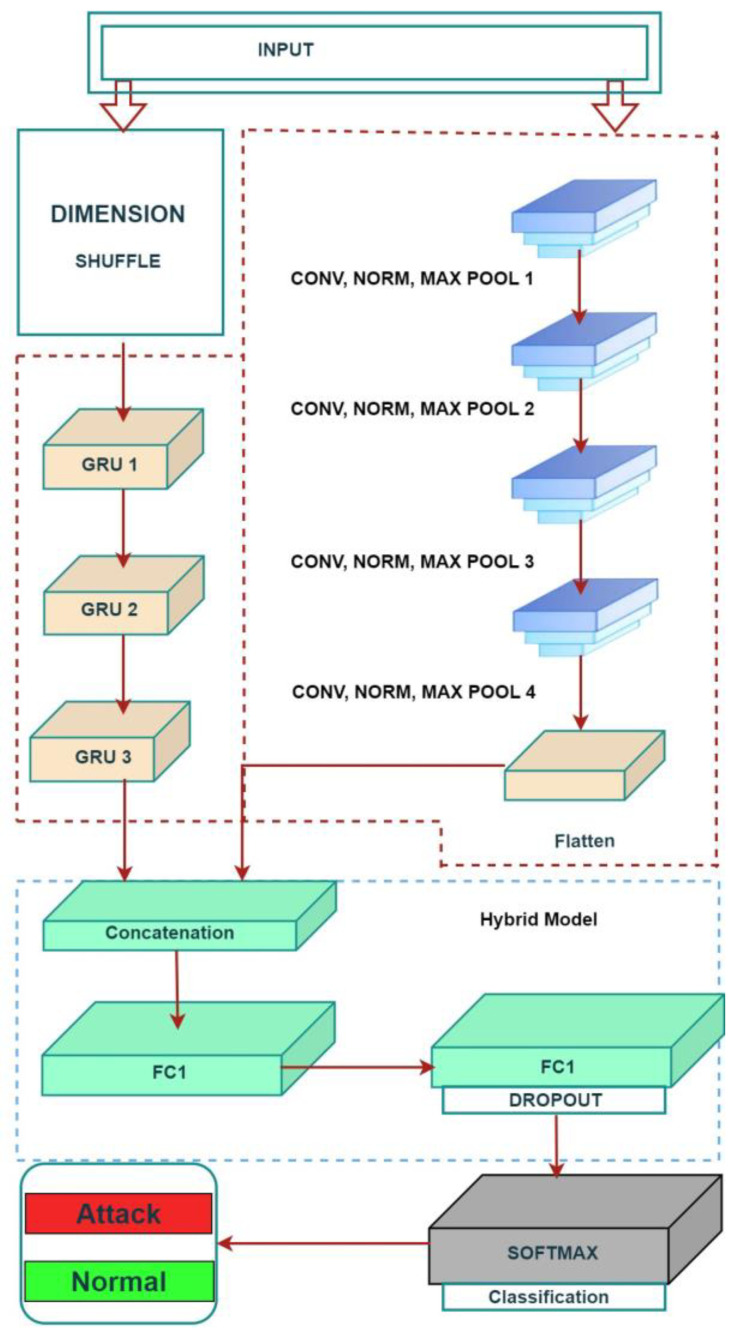
Proposed CNN–GRU hybrid model.

**Figure 4 sensors-23-07464-f004:**
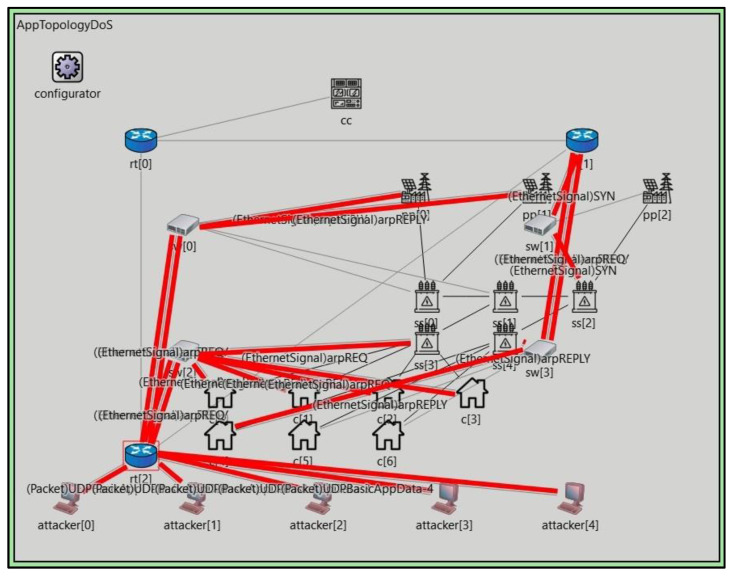
Omnet++ experiment simulation.

**Figure 5 sensors-23-07464-f005:**
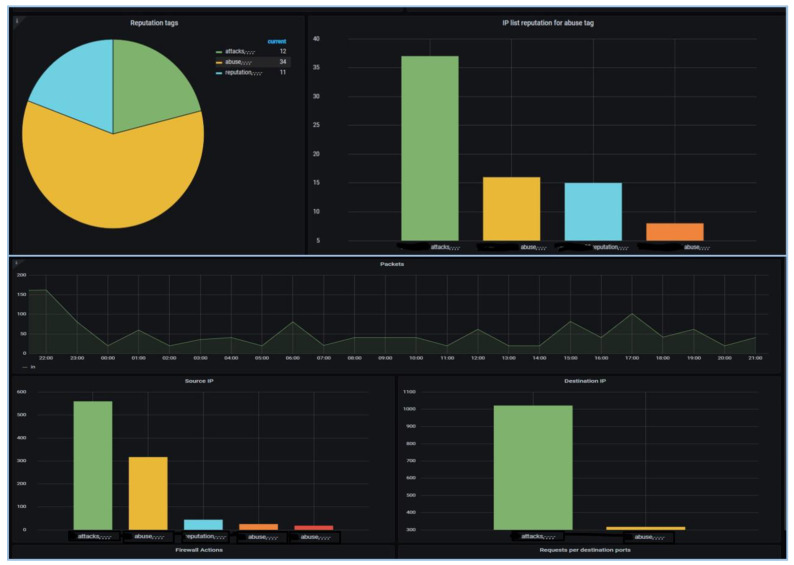
Grafana Smart Grid IDS dashboard.

**Figure 6 sensors-23-07464-f006:**
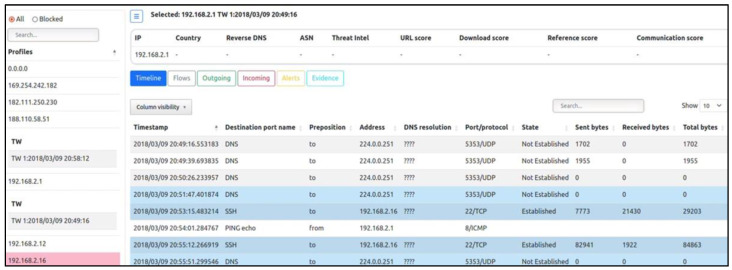
Slips integrate with CNN and GRU for real-time detection.

**Figure 7 sensors-23-07464-f007:**
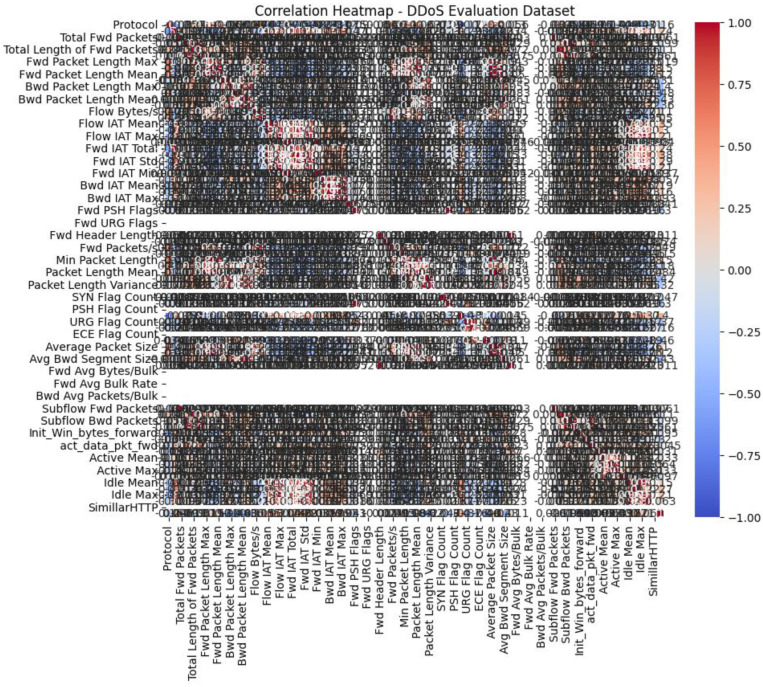
Heatmap diagram.

**Figure 8 sensors-23-07464-f008:**
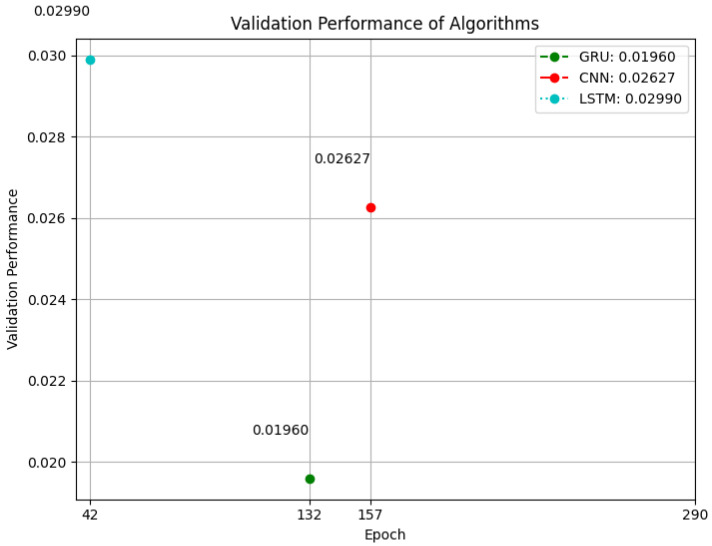
Validation Performance of Algorithms.

**Figure 9 sensors-23-07464-f009:**
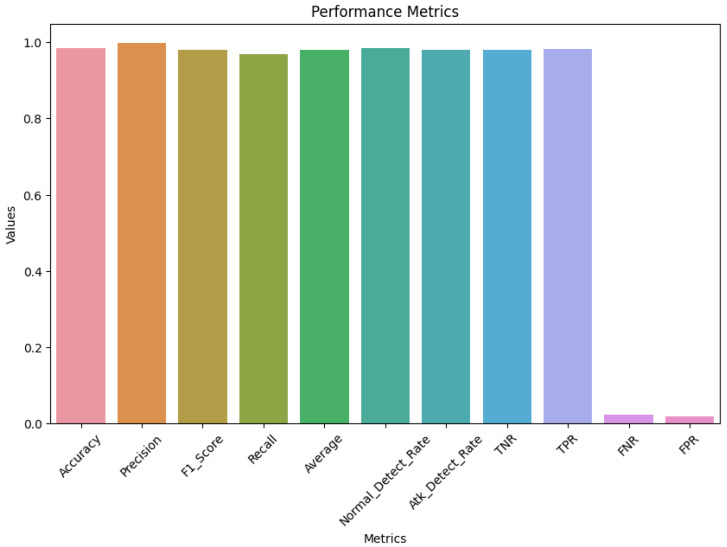
CNN and hybrid model results.

**Figure 10 sensors-23-07464-f010:**
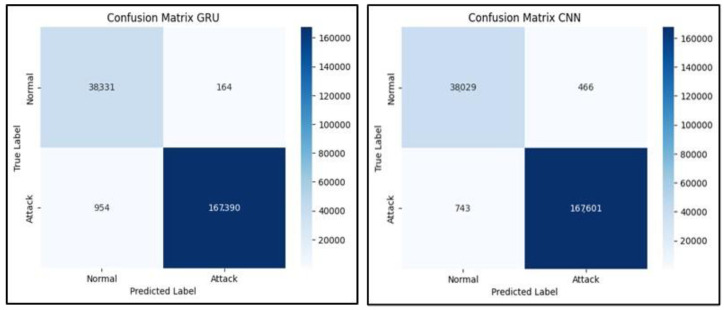
CNN and GRU matrix.

**Figure 11 sensors-23-07464-f011:**
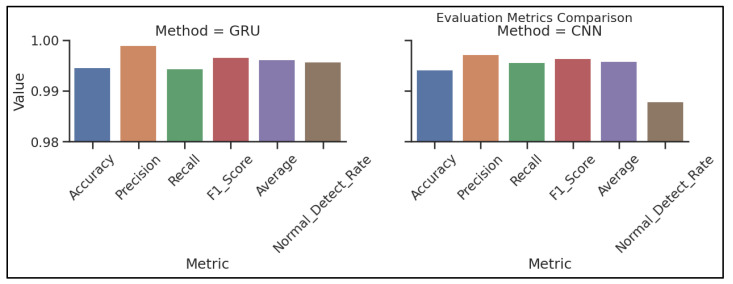
Performance comparison of individual models.

**Figure 12 sensors-23-07464-f012:**
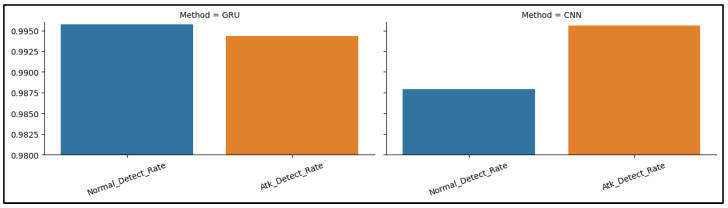
CNN and GRU attack and normal detection rate.

**Figure 13 sensors-23-07464-f013:**
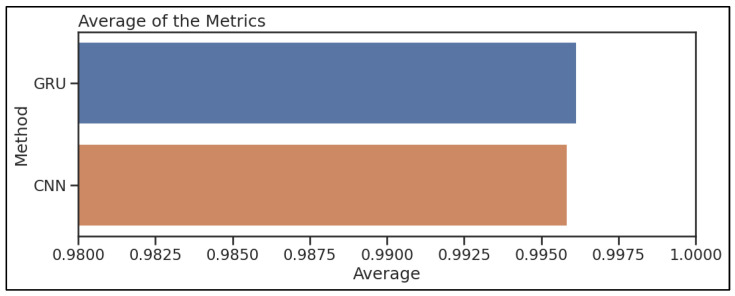
CNN and GRU average results.

**Table 1 sensors-23-07464-t001:** Dataset characteristics for the testbed.

Attacks Type	Records	Training Records	Test Records	Label
BENIGN	133,795	106,874	26,921	0
DDoS	1,311,770	997,054	314,716	1

**Table 2 sensors-23-07464-t002:** Hyperparameter configuration for CNN−GRU.

No	Hyperparameter	Recommended Values
1	Learning Rate	0.001
2	Number of Convolutional Layers	2–3
3	Number of GRU Units	32, 64, 128
4	Dropout Rate	0.2–0.5
5	Batch Size	16–128
6	Number of Epochs	50–100

**Table 3 sensors-23-07464-t003:** Hyperparameter configuration for CNN−GRU.

Paper	Models	True Positive Rate %	Accuracy %	F1Score %	False Positive %
[38]	RNN	-	91.97	91.20	8.00
[36]	Ensemble	-	97.50	97.20	3.45
Our	Hybrid	100	99.86	99.68	0.22

## Data Availability

Not applicable.

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
