# Peer review of "Ensemble Model Based on Hybrid Deep Learning for Intrusion Detection in Smart Grid Networks"

_sensors, 2023, doi:10.3390/s23177464_

Round 1
Reviewer 1 Report
The paper proposes a hybrid deep learning method for detecting distributed denial-of-service attacks on the Smart Grid's communication infrastructure. The method combines the convolutional neural network and gated recurrent unit algorithms. The comments are as follows:
1. Please provide the implementation details of the experiments, including the experiment environment, how to determine the number of Convolutional Layers and GRU Units.
2. Section 2 is too long and unclear. Please divide it into sub-sections.
3. Please correct the format problem in Figure 8. There is data overlap in the display.
4. Please provide an ablation study to demonstrate the effectiveness of the modules in the proposed CNN-GRU hybrid model.
5. In the experiments, please compare the proposed method with other state-of-the-art methods, if possible.
Moderate editing of English language required
Reviewer 2 Report
1.The paper is well written,and contributes a hybrid deep learning approach for detecting distributed denial-of-service attacks on the Smart Grid’s communication infrastructure,which enables accurately identifying malicious activities and providing reliable security measures for the smart enery grid.And the proposed can achieve higher accuracy.
2. There are some problems, which must be solved before it is considered for publication. If the following problems are well-addressed, this reviewer believes that the essential contribution of this paper are important for Smart Grid.
(1)The article is illustrated with vague pictures,such as Figure 1,Figure 4 and Figure 6.
(2)Line 413 on page 9,the matrix transpose is formatted incorrectly.
(3)Are formulas 15 and 16 formatted incorrectly?
(4)Lines 683-696 are cluttered.
(5)The format of the headings in Table 2 and Figure 12 is inconsistent with the others.
(6)Chapter 2 only lists relevant work, without comparing this paper with previous work, and does not explain the innovation and improvement of this paper compared with previous articles. It is necessary to improve the background introduction and make a general introduction to the current development level of this issue.
Reviewer 3 Report
This paper proposes a hybrid deep learning approach specifically designed for detecting distributed denial-of-service attacks on the Smart Grid's communication infrastructure.
1. In the abstract, it is written "Experimental results demonstrate....". Do they were performed or only simulations? Similarly, section 5 needs to be renamed.
2. The abbreviations need to be defined the first time and later used as it is. There is no need to define these again and again.
3. The quality of Figure 1 needs to be improved.
4. Define all the parameters in equations 12,13 etc.
5. The future works section is missing from the conclusion.
A few minor edits are required.
Round 2
Reviewer 1 Report
Recommend to accept this paper based on the responses to the reviews in previous cycle.